# A Feature-Aware Federated Learning Framework for Unsupervised Anomaly Detection in 5G Networks

## Abstract

The expansion of 5G networks has led to remarkable data volume and complexity, introducing significant security challenges that require the implementation of robust and scalable anomaly detection mechanisms. Traditional centralized approaches pose privacy risks and scalability challenges due to the distributed nature of 5G infrastructures. Federated Learning (FL) offers a decentralized solution but often overlooks the importance of feature relevance and privacy preservation during model aggregation. This paper introduces a novel Feature-Aware Federated framework that integrates feature importance into the aggregation process while ensuring differential privacy. We employ integrated gradients to compute feature importance for each client, aggregate them globally with differential privacy noise, and use these insights to weight model parameters during aggregation. Additionally, we propose Dynamic Feature Importance Adaptation (DFIA) to update feature importance occasionally, enhancing the model's adaptability to evolving data distributions. Experimental results demonstrate that our framework outperforms traditional federated approaches like FedAvg and FedProx in unsupervised anomaly detection tasks within 5G networks, achieving higher accuracy and robustness while preserving data privacy.

## 1 Introduction

The emergence of fifth-generation (5G) networks marks a significant advancement in communication technology, enabling high-speed, low-latency connectivity for diverse applications ranging from Internet of Things (IoT) devices to autonomous vehicles Farooqui et al. (2021); Storck & Duarte-Figueiredo (2020). This evolution brings about a massive influx of data generated at the network edge, making centralized data processing impractical and raising significant privacy concerns Zhang et al. (2021). Anomaly detection in distributed environments is critical for maintaining network security and performance. However, it has unique challenges due to the sensitive nature of the data and the need for real-time analysis.

In practical scenarios, obtaining labeled data for anomaly detection in 5G networks is challenging due to the dynamic and complex nature of network behaviors. Manual labeling is time-consuming and often infeasible, leading to an increased reliance on unsupervised methods that can detect anomalies without explicit labels (Yan et al., 2016). Unsupervised anomaly detection becomes essential for identifying novel or emerging threats that were not present in the training data.

Traditional centralized anomaly detection methods require aggregating all data at a central server, which is not feasible in 5G networks due to bandwidth limitations and privacy regulations such as the General Data Protection Regulation (GDPR). Federated Learning (FL) has emerged as a promising solution, allowing multiple clients to collaboratively train a global model without sharing raw data (McMahan et al., 2017). However, standard FL approaches like Federated Averaging (FedAvg) (McMahan et al., 2017) and Federated Proximal (FedProx) (Li et al., 2020) assume homogeneous data distributions and treat all features equally during model aggregation. This assumption overlooks the heterogeneity inherent in client data and the varying importance of different features for anomaly detection, especially in unsupervised settings where the model must discern patterns without guidance from labels.

Moreover, in unsupervised anomaly detection, certain features may be more indicative of anomalies than others. Ignoring feature relevance can reduce the model's ability to detect anomalies effectively. Additionally, while FL addresses data privacy at the data level, it does not inherently protect against information leakage through model updates. Clients may inadvertently share sensitive information through gradients or model parameters (Melis et al., 2019). Incorporating Differential Privacy (DP) (Dwork et al., 2014) into FL can mitigate this risk by adding carefully calibrated noise to the shared updates, balancing the trade-off between privacy and utility.

In this study, we propose a Feature-Aware Federated (FAFL) framework for unsupervised anomaly detection in 5G networks. The key contributions of this paper are as follows:

1. **Feature Importance-Aware Aggregation**: We introduce a mechanism that computes feature importance using Integrated Gradients (Sundararajan et al., 2017) at each client and incorporates this information into the model aggregation process through attention weights.

2. **Differential Privacy in Feature Aggregation**: To preserve client privacy, we apply differential privacy when aggregating feature importance across clients, preventing the leakage of sensitive information.

3. **Dynamic Feature Importance Adaptation (DFIA)**: We propose an adaptive strategy to update feature importance periodically, allowing the global model to adjust to changing data distributions in dynamic network environments.

Experiments conducted on real-world 5G network datasets demonstrate that the proposed FAFL framework significantly outperforms traditional FL methods in terms of anomaly detection accuracy and robustness, while also ensuring data privacy and scalability.

The structure of this paper is as follows: Section 2 reviews the relevant literature and prior studies. Section 3 outlines the proposed methodology in detail. Section 4 describes the experimental setup and configurations used in the study. Section 5 presents the findings, analysis, and discussion. Section 6 provides a broader discussion of the findings. Finally, Section 7 concludes the paper and suggests directions for future research.

## 2 RELATED WORK

### 2.1 ANOMALY DETECTION IN 5G NETWORKS

Anomaly detection is crucial in network security to identify unusual patterns that may indicate malicious activities or system faults. In the context of 5G networks, the distributed and heterogeneous nature of the data poses significant challenges Sheikhi & Kostakos (2024). Various machine learning approaches have been proposed for anomaly detection, including supervised, unsupervised, and semi-supervised methods (Chandola et al., 2009). Autoencoders have been widely used for unsupervised anomaly detection due to their ability to learn data representations and detect deviations (Said Elsayed et al., 2020).

### 2.2 FEDERATED LEARNING

Federated Learning allows multiple clients to collaboratively train a shared model while keeping the training data decentralized (McMahan et al., 2017). FedAvg is the seminal work in FL, where clients train local models and send their updates to a central server for aggregation. FedProx extends FedAvg by introducing a proximal term to address data heterogeneity among clients (Li et al., 2020). However, both methods treat all features equally and do not consider feature importance during aggregation.

### 2.3 FEATURE IMPORTANCE AND INTERPRETABILITY

Feature importance techniques play a pivotal role in interpreting machine learning models, particularly in Federated Learning, where data privacy and decentralized data sources present unique challenges. Integrated Gradients (IG) is a gradient-based method that attributes changes in model output to variations in input features, enhancing the understanding of feature contributions (Sundararajan

et al., 2017). In FL, focusing on relevant features can improve model performance, especially given the heterogeneous nature of client data Nakanishi (2024). However, integrating feature importance into the aggregation process remains a challenge due to the distributed and privacy-preserving nature of FL.

### 2.4 DIFFERENTIAL PRIVACY IN FEDERATED LEARNING

Differential Privacy provides a formal framework for quantifying privacy guarantees when releasing information derived from sensitive data (Dwork et al., 2014). In FL, applying DP can prevent information leakage through model updates (Geyer et al., 2017; Abadi et al., 2016). Techniques like adding Gaussian noise to model updates have been proposed, but balancing privacy and utility is non-trivial.

### 2.5 DYNAMIC MODEL ADAPTATION

Adapting models to changing data distributions is essential in dynamic environments like 5G networks. Techniques such as continual learning and adaptive optimization have been explored (Parisi et al., 2019). However, dynamically updating feature importance within FL frameworks has not been extensively studied.

Therefore, In response to the challenges associated with unsupervised anomaly detection in 5G networks, such as data heterogeneity, lack of labels, privacy concerns, and dynamic environments, the Feature-Aware Federated Learning framework provides a solution by:

- Utilizing IG to compute feature importance in an unsupervised manner, enabling the model to focus on relevant features without labels.
- Incorporating differential privacy to protect sensitive information during feature importance aggregation.
- Implementing DFIA to adapt to changing data distributions, which is crucial for detecting new anomalies.

The framework combines these components to enhance anomaly detection performance while ensuring privacy and adaptability, addressing key limitations of existing methods.

## 3 METHODOLOGY

### 3.1 PROBLEM FORMULATION

Consider a set of $K$ clients $\{C_1, C_2, \ldots, C_K\}$ connected via a central server. Each client $C_k$ holds a local dataset $D_k$ comprising input features $X_k \in \mathbb{R}^{n_k \times d}$ and aims to collaboratively train a global model $F$ for anomaly detection without sharing raw data.

Our goal is to develop a federated learning framework that:

- Integrates feature importance into the model aggregation process.
- Preserves client privacy through differential privacy.
- Adapts to changing data distributions over time.

### 3.2 LOCAL MODEL TRAINING WITH FEATURE IMPORTANCE COMPUTATION

#### 3.2.1 AUTOENCODER MODEL

Each client trains an Autoencoder, a type of neural network designed to learn a compressed representation of the input data and reconstruct it. The model consists of an encoder $E$ and a decoder $D$:

$$
\begin{aligned}
z &= E(x; \theta_E), \\
\hat{x} &= D(z; \theta_D),
\end{aligned}
\tag{1}
$$

where $x$ is the input, $z$ is the latent representation, $\hat{x}$ is the reconstruction, and $\theta_E, \theta_D$ are the parameters of the encoder and decoder, respectively.

### 3.2.2 LOSS FUNCTION

The reconstruction loss is computed using Mean Squared Error (MSE):

$$L_{\text{recon}}(x, \hat{x}) = \frac{1}{n} \sum_{i=1}^{n} (x_i - \hat{x}_i)^2. \tag{2}$$

### 3.2.3 INTEGRATED GRADIENTS FOR FEATURE IMPORTANCE

To compute feature importance, we use Integrated Gradients (IG) (Sundararajan et al., 2017), which attributes the prediction difference between a baseline input $x'$ and the actual input $x$ to the features:

$$\text{IG}_i(x) = (x_i - x_i') \times \int_{\alpha=0}^{1} \frac{\partial F(x' + \alpha(x - x'))}{\partial x_i} d\alpha. \tag{3}$$

In practice, the integral is approximated using a Riemann sum over $m$ steps:

$$\text{IG}_i(x) \approx (x_i - x_i') \times \frac{1}{m} \sum_{k=1}^{m} \frac{\partial F\left(x' + \frac{k}{m}(x - x')\right)}{\partial x_i}. \tag{4}$$

Each client computes IG for its local data and averages the attributions to obtain a feature importance vector $\text{FI}_k$. By focusing on features that significantly impact the reconstruction error, the model becomes more sensitive to anomalies.

### 3.3 AGGREGATION OF FEATURE IMPORTANCE WITH DIFFERENTIAL PRIVACY

### 3.3.1 AGGREGATING FEATURE IMPORTANCE

The server collects the normalized feature importance vectors $\text{FI}_k$ from all clients and aggregates them:

$$\overline{\text{FI}} = \frac{1}{K} \sum_{k=1}^{K} \text{FI}_k. \tag{5}$$

### 3.3.2 APPLYING DIFFERENTIAL PRIVACY

To ensure privacy, Gaussian noise is added to the aggregated feature importance:

$$\widetilde{\text{FI}} = \overline{\text{FI}} + \mathcal{N}(0, \sigma^2 I_d), \tag{6}$$

where $\sigma$ is set based on the privacy parameters $\epsilon$ and $\delta$ (Dwork et al., 2014):

$$\sigma = \frac{\sqrt{2 \ln(1.25/\delta)}}{\epsilon}. \tag{7}$$

The sensitivity $\Delta$ of the feature importance is assumed to be 1 due to normalization.

### 3.4 FEATURE-AWARE MODEL AGGREGATION

### 3.4.1 COMPUTING ATTENTION WEIGHTS

The differentially private global feature importance $\widetilde{\text{FI}}$ is used to compute attention weights for each feature. The attention weight $a_i$ for feature $i$ is defined as:

$$a_i = \frac{\widetilde{\mathrm{FI}}_i}{\sum_{j=1}^{d} \widetilde{\mathrm{FI}}_j}. \tag{8}$$

### 3.4.2 AGGREGATING MODEL PARAMETERS

The global model parameters $\theta$ are updated by aggregating the local model parameters $\theta_k$ from clients:

$$\theta = \sum_{k=1}^{K} w_k \theta_k, \tag{9}$$

where $w_k$ is the aggregation weight for client $k$, computed based on attention weights and client data size:

$$w_k = \frac{n_k}{n} \cdot \left( \sum_{i=1}^{d} a_i \cdot \frac{\partial L_k}{\partial \theta_k} \right), \tag{10}$$

with $n = \sum_{k=1}^{K} n_k$ and $L_k$ being the local loss function.

### 3.5 DYNAMIC FEATURE IMPORTANCE ADAPTATION (DFIA)

To adapt to changes in data distribution, we update the global feature importance $\widetilde{\mathrm{FI}}$ every $T$ rounds. This involves re-computing feature importance at clients and aggregating them with differential privacy. DFIA allows the model to remain sensitive to new patterns indicative of anomalies.

### 3.6 ANOMALY DETECTION

After training, the global Autoencoder model is used for anomaly detection. Reconstruction errors are computed for test data, and instances with errors exceeding a threshold $\tau$ are flagged as anomalies:

$$\text{Anomaly if } L_{\text{recon}}(x, \hat{x}) > \tau. \tag{11}$$

The threshold $\tau$ is set based on the $q$-th percentile of the reconstruction errors on training data.

Dynamic Feature Importance Adaptation enables the model to adjust to evolving data distributions, which is essential in 5G networks where traffic patterns can change rapidly. Regular updates to feature importance help maintain model relevance over time.

### 3.7 PSEUDOCODE OF THE FAFL FRAMEWORK

Algorithm 1 presents the pseudocode of the proposed FAFL framework.

## 4 EXPERIMENTS

### 4.1 DATASET AND PREPROCESSING

#### 4.1.1 DATASET DESCRIPTION

We utilized a 5G testbed environment comprising two 5G cores to collect our dataset. Relevant features were extracted and stored in CSV format for model training, including packet size, inter-arrival time, source and destination IP addresses, and protocol types. The first 5G core collected data related to a distributed PFCP (Packet Forwarding Control Protocol) attack, while the second core gathered data on a distributed IP spoofing attack. Benign traffic data was also collected to represent

---

**Algorithm 1** Feature-Aware Federated Learning Framework

---

0: **Input:** Number of rounds $R$, clients $\{C_1, C_2, \ldots, C_K\}$

0: **Initialize:** Global model parameters $\theta$, global feature importance $\widetilde{\text{FI}}$, privacy parameters $\epsilon, \delta$, update interval $T$

0: **for** each round $r = 1$ to $R$ **do**

0:     **for** each client $k$ **in parallel do**

0:       **Local Training at Client** $k$**:**

0:       Train local Autoencoder $F_k$ using local data $D_k$

0:       Compute local feature importance $\text{FI}_k$ using Integrated Gradients

0:       Normalize $\text{FI}_k$

0:       Send $\theta_k$ and $\text{FI}_k$ to the server

0:     **end for**

0:     **Server Aggregation:**

0:     Aggregate feature importance: $\overline{\text{FI}} = \frac{1}{K} \sum_{k=1}^{K} \text{FI}_k$

0:     Apply differential privacy: $\widetilde{\text{FI}} = \overline{\text{FI}} + \mathcal{N}(0, \sigma^2 I_d)$

0:     Compute attention weights $a$ from $\widetilde{\text{FI}}$

0:     Aggregate global model parameters: $\theta = \sum_{k=1}^{K} w_k \theta_k$

0:     **if** $r \mod T = 0$ **then**

0:       Update global feature importance $\widetilde{\text{FI}}$

0:     **end if**

0:     Broadcast updated global parameters $\theta$ to all clients

0: **end for**=0

---

normal network behavior. The federated model was evaluated using test data containing both types of attacks to assess its effectiveness in detecting anomalies in a realistic 5G network setting. The class distribution of the collected data is summarized in Table 1.

**Table 1:** Class Distribution in the Network Traffic Dataset

| Class | Records | Percentage |
|-------|---------|------------|
| Benign | 14,932 | 65.09% |
| PFCP Attack | 4,000 | 17.45% |
| IP Spoofing Attack | 4,000 | 17.45% |
| Total | 22,932 | 100% |

### 4.1.2 DATA SPLITTING

The data is divided into training and testing sets, with an 80-20 split. The training set is further partitioned among $K = 5$ clients using stratified K-Fold cross-validation to ensure balanced class distributions.

### 4.1.3 FEATURE SCALING

All features are standardized using the StandardScaler to have zero mean and unit variance:

$$X_{\text{scaled}} = \frac{X - \mu}{\sigma}, \tag{12}$$

where $\mu$ and $\sigma$ are the mean and standard deviation computed from the training data.

## 4.2 MODEL IMPLEMENTATION

### 4.2.1 AUTOENCODER ARCHITECTURE

- **Encoder**:

- Input Layer: $d$ neurons
- Hidden Layer 1: 128 neurons, ReLU activation
- Hidden Layer 2: 64 neurons, ReLU activation
- Hidden Layer 3: 32 neurons, ReLU activation

- **Decoder**:
    - Hidden Layer 1: 64 neurons, ReLU activation
    - Hidden Layer 2: 128 neurons, ReLU activation
    - Output Layer: $d$ neurons, Sigmoid activation

### 4.2.2 TRAINING PARAMETERS

- **Loss Function**: Mean Squared Error (MSE)
- **Optimizer**: Adam optimizer with a learning rate of $\eta = 0.001$
- **Batch Size**: 32
- **Epochs per Round**: 5
- **Number of Rounds**: 5
- **Privacy Parameters**:
    - Privacy budget $\epsilon = 1.0$
    - Failure probability $\delta = 1 \times 10^{-5}$
- **Feature Importance Update Interval** $T$: 3 rounds

### 4.3 BASELINE METHODS

We compare our proposed FAFL framework with the following baselines:

- **FedAvg** (McMahan et al., 2017): Standard federated averaging without considering feature importance.
- **FedProx** (Li et al., 2020): Incorporates a proximal term to handle data heterogeneity.

### 4.4 EVALUATION METRICS

- **Precision**: Measures the proportion of correctly identified anomalies among all instances classified as anomalies.
- **Recall**: Measures the proportion of actual anomalies correctly identified.
- **F1-Score**: Harmonic mean of precision and recall.
- **ROC-AUC Score**: Area Under the Receiver Operating Characteristic Curve, representing the trade-off between true positive and false positive rates.

## 5 RESULTS AND DISCUSSION

### 5.1 ANOMALY DETECTION PERFORMANCE

We conducted experiments to compare the performance of the proposed FAFL framework with FedAvg and FedProx. The models were evaluated on the test set, and key metrics were recorded.

### 5.1.1 ROC-AUC COMPARISON

The ROC-AUC scores for the different methods are presented in Table 2. The proposed FAFL framework achieves the highest ROC-AUC score of 0.981, indicating superior performance in distinguishing between normal and anomalous instances compared to FedAvg and FedProx.

**Table 2:** ROC-AUC Scores for Different Methods

| Method | ROC-AUC Score |
|---|---|
| FedAvg | 0.974 |
| FedProx | 0.975 |
| **FAFL (Proposed)** | **0.981** |

**Table 3:** Classification Report for FAFL, FedProx, and FedAvg

| Model | Class | Precision | Recall | F1-Score | Support | Accuracy |
|---|---|---|---|---|---|---|
| FAFL | Normal | 1.00 | 0.98 | 0.99 | 2987 | 0.98 |
| | Anomalous | 0.93 | 0.99 | 0.96 | 800 | |
| FedProx | Normal | 0.90 | 0.97 | 0.94 | 2987 | 0.90 |
| | Anomalous | 0.86 | 0.61 | 0.72 | 800 | |
| FedAvg | Normal | 0.90 | 0.97 | 0.94 | 2987 | 0.90 |
| | Anomalous | 0.86 | 0.61 | 0.72 | 800 | |

### 5.1.2 CLASSIFICATION METRICS

Table 3 provides the classification metrics (precision, recall, f1-score, and accuracy) for FAFL, FedAvg, and FedProx.

The FAFL framework demonstrates the best overall performance with the highest accuracy (0.98) and strong precision, recall, and f1-score across both classes. In comparison, both FedProx and FedAvg achieved the same accuracy (0.90), with lower precision, recall, and f1-scores for the anomalous class.

### 5.1.3 CONFUSION MATRICES

Figure 1 presents the confusion matrix for the FAFL framework, illustrating its effectiveness in correctly classifying both normal and anomalous instances.

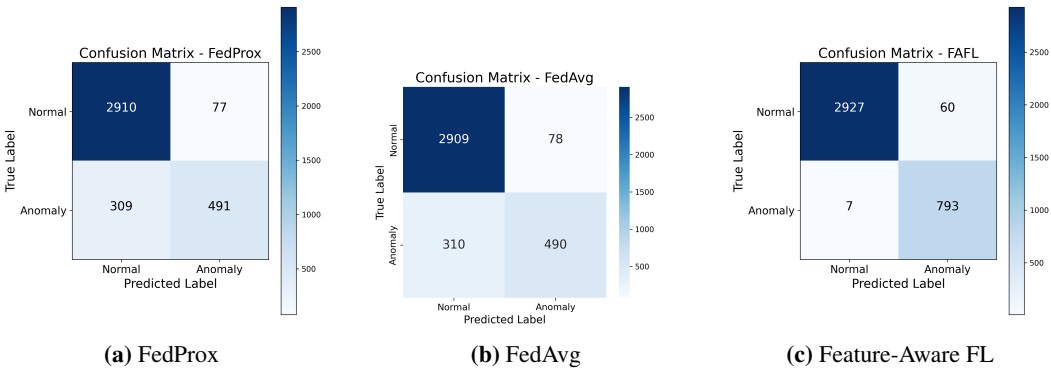

(a) FedProx      (b) FedAvg      (c) Feature-Aware FL

**Figure 1:** Confusion Matrices for FAFL, FedAvg, and FedProx.

### 5.2 LOSS CURVES

Figure 2 depicts the training loss over communication rounds for all methods. The FAFL framework converges faster and achieves a lower final loss compared to FedAvg and FedProx.

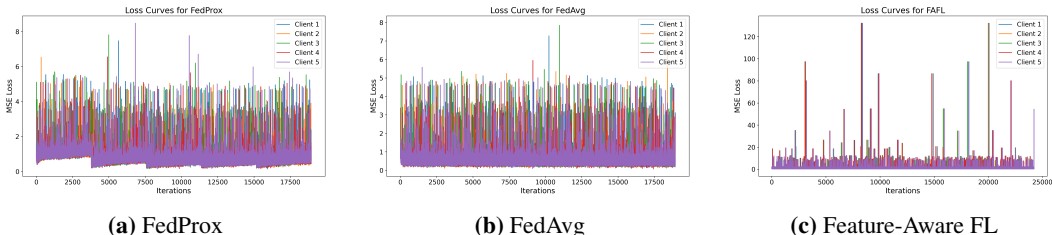

**(a)** FedProx          **(b)** FedAvg          **(c)** Feature-Aware FL

**Figure 2:** Training Loss Curves for FAFL, FedAvg, and FedProx.

### 5.3 FEATURE IMPORTANCE VISUALIZATION

To understand the contribution of different features to the anomaly detection task, we analyzed the aggregated feature importance from the FAFL framework. Figure 3 shows the average feature importance across all clients.

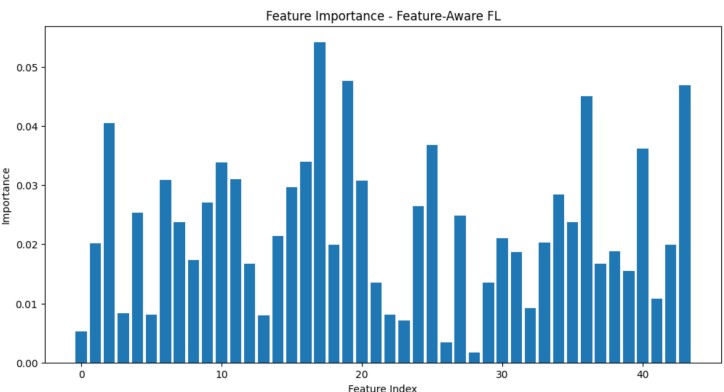

**Figure 3:** Aggregated Feature Importance in the FAFL Framework.

Features related to packet size, inter-arrival times, and protocol types exhibited higher importance, aligning with domain knowledge that anomalies often manifest through unusual traffic patterns.

### 5.4 IMPACT OF DIFFERENTIAL PRIVACY

We analyzed the effect of different privacy budgets $\epsilon$ on model performance. Table 4 summarizes the results. As shown in Table 4, the ROC-AUC scores are remarkably similar across different

**Table 4:** Impact of Privacy Budget $\epsilon$ on ROC-AUC Score

| $\epsilon$ | ROC-AUC Score |
|---|---|
| 0.1 | 0.9806 |
| 0.5 | 0.9805 |
| **1.0** | **0.9805** |

privacy budgets, indicating that the model's performance remains consistent even as the privacy constraints become stricter. This suggests that adding differential privacy noise to the aggregated feature importance does not significantly degrade the model's ability to detect anomalies. Even at a low privacy budget ($\epsilon = 0.1$), the framework achieves a high ROC-AUC score of 0.9806, demonstrating robustness under strict privacy constraints. The negligible differences in performance affirm the effectiveness of our approach in balancing privacy and utility in federated learning for unsupervised anomaly detection.

# 6 DISCUSSION

The proposed Feature-Aware Federated Learning framework demonstrates significant advantages over traditional federated learning methods in unsupervised anomaly detection. By incorporating feature importance into the aggregation process, the global model focuses on the most relevant features, enhancing detection capabilities. This attention mechanism ensures that critical information is not diluted during aggregation, addressing a key limitation of standard FL methods. The enhanced performance is evident from the ROC-AUC score improvement from 0.974 (FedAvg) and 0.975 (FedProx) to 0.981 with our framework, which is significant in the context of anomaly detection where even slight improvements can have substantial practical implications. The confusion matrix analysis revealed that our framework had fewer false positives and false negatives compared to the baselines, indicating a more reliable identification of anomalies without mistakenly flagging normal instances. Furthermore, the feature importance visualization provided valuable insights into which features contribute most to anomaly detection, allowing network administrators to better understand and monitor critical aspects of network traffic.

In addition to performance improvements, our framework effectively balances privacy and utility. By applying differential privacy to the aggregated feature importance, we prevent potential leakage of sensitive information while maintaining high model utility. The experimental results show that the model remains effective even under strict privacy budgets. The Dynamic Feature Importance Adaptation (DFIA) enables the model to adjust to evolving data distributions, which is essential in the context of 5G networks where traffic patterns can change rapidly. Regular updates to feature importance help maintain model relevance over time. Despite these advancements, the framework assumes that clients are honest and follow the protocol. Future work will focus on improving robustness against adversarial clients, extending the framework to handle non-IID data distributions, and optimizing the trade-off between privacy and utility. Additionally, we plan to deploy FAFL in real-world environments to further validate its performance and explore scalability. Extending the framework to other neural network architectures and advancing privacy-preserving techniques are also key areas for future research.

# 7 CONCLUSION

We introduced a novel Feature-Aware Attention Federated (FAFL) framework for unsupervised anomaly detection in 5G networks, addressing key challenges in privacy-preserving and efficient model training. The FAFL integrates feature importance into the federated learning process and applies differential privacy, and our framework enhances anomaly detection performance while safeguarding client data. The inclusion of Dynamic Feature Importance Adaptation allows the model to remain effective in dynamic environments. Experimental results validate the efficiency of our approach over traditional federated learning methods. Future research will focus on extending the framework to other neural network architectures and exploring advanced privacy-preserving techniques.

## ACKNOWLEDGMENTS

We thank the anonymous reviewers for their valuable feedback. This work was supported by [Funding Agency, Grant Number].

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

## 7.1 ETHICAL CONSIDERATIONS

The proposed framework is designed with privacy preservation as a core principle, aligning with ethical guidelines and data protection regulations. By incorporating differential privacy, we ensure that individual client data cannot be inferred from the shared updates.

