# OpenReview forum: "A Feature-Aware Federated Learning Framework for Unsupervised Anomaly Detection in 5G Networks"
_ICLR.cc/2025/Conference — ICLR 2025 Conference Withdrawn Submission_

### Official Review · Reviewer_4wNZ · 2024-10-23

**Soundness:** 1
**Presentation:** 2
**Contribution:** 1
**Rating:** 1
**Confidence:** 5

**Summary:**

The authors propose an explainable AI-based algorithm, integrated gradients, to employ the feature importance of each client in the aggregation of the FL system, apply it to the 5G-related tasks, and process the importance estimation in each round to adopt the dynamic feature of 5G networks. Results compared with traditional aggregation algorithms, such as FedAvg and FedProx show some performance gain on one standard dataset.

**Strengths:**

1. The method of employing integrated gradients to estimate the contribution/importance of each client is interesting.

**Weaknesses:**

1. The proposed scheme seems standard and simply applied to the FL system. Details on the key steps are largely omitted. For example, it is not clear how the local client calculates the feature importance of its own data. According to Eq. 3 and my basic understanding of integrated gradients, this step needs a basis from other clients or the global model. However, this detail has not been fully explored in the provided content.

2. To adopt the dynamic characters of the environment, the authors just let the importance calculation happen in every T round, which is kind of straight and trivial. In addition, the advantages of this step are not investigated in the experiments.

3. Wrong use of DP. Adding random noise on the feature importance will not protect the raw data of clients, and the related experiments are also confusing as many details are not provided.

**Questions:**

1. How to estimate the feature importance without a basis from other participants?

2. What is the reason that the client does the importance estimation itself? What if any malicious or dishonest client exists in the system?

3. What are the benefits of adding noises to the importance?

4. Many key steps or designs lack technical details, is there any specific reason for this?

---

### Official Review · Reviewer_nMd9 · 2024-10-29

**Soundness:** 3
**Presentation:** 2
**Contribution:** 1
**Rating:** 3
**Confidence:** 4

**Summary:**

This paper proposes a Feature-Aware Federated Learning (FAFL) framework, which incorporates feature importance into FL process. The framework utilizes integrated gradients to compute feature relevance and adapts dynamically to changing data distributions, aiming to handle the heterogeneity of client data.

**Strengths:**

The modeling of the 5G scenario in the paper is commendable, and the proposed method is an interesting attempt to handle heterogeneous features.

**Weaknesses:**

1. The background research in this paper is insufficient. There is already a substantial body of work focused on heterogeneous data and features in FL, but the authors only mention the classic approaches, FedAvg and FedProx.

2. The experiments are lacking. The authors only use the two aforementioned methods as baselines and conduct a limited number of experiments. A significant portion of the paper is devoted to details about model architecture.

3. The paper lacks motivation. Why do we need a feature weighting method instead of letting the model learn it? It would be better if the necessity of this method in certain scenarios could be clarified.

4. In the introduction, it’s stated: "this (FedAvg’s) assumption overlooks the heterogeneity inherent in client data," which suggests that the 5G scenario involves non-i.i.d data. However, the authors' method directly averages the 'feature importance vectors' from different clients (Eq.5). The rationale behind this step doesn’t seem intuitive.

5. Eq.10 seems incorrect. Since \theta_k is a vector representing the model parameters, this implies that w_k should also be a vector. Meanwhile, \Sigma a_i doesn’t appear to make sense. The unclear nature of this formula makes it difficult to fully understand the proposed method.

**Questions:**

See weaknesses.

---

### Official Review · Reviewer_XkFq · 2024-10-31

**Soundness:** 2
**Presentation:** 1
**Contribution:** 2
**Rating:** 3
**Confidence:** 4

**Summary:**

Federated learning provides a decentralised solution for anomaly detection in 5G networks, but neglects the importance of feature relevance and privacy preservation during model aggregation. This paper presents a federated learning framework for 5G networks, incorporating feature importance to improve anomaly detection. It uses integrated gradients to determine feature relevance and dynamically adapts to changing data distributions, integrating differential privacy to secure data. Experiments on real-world 5G network datasets show the advantages of the FAFL framework in terms of anomaly detection accuracy and robustness.

**Strengths:**

1、The proposed framework effectively addresses conditional joint learning methods in the context of data heterogeneity, lack of labelling, privacy issues and dynamic environment challenges faced in 5G network scenarios.
2、The paper considers the protection of customer privacy by applying differential privacy when aggregating cross-customer feature importance to prevent leakage of sensitive information.
3、The paper utilises a 5G testbed environment containing two 5G cores to collect datasets and conduct experiments with a certain degree of authenticity and credibility that are relevant.

**Weaknesses:**

1、The applicable scenario of the proposed method is unclear. Further clarification is necessary.
2、The paper lacks innovation, appearing as a fusion of multiple existing technologies rather than presenting original contributions.
3、This paper compares a limited number of methods, necessitating the inclusion of additional federated anomaly detection techniques to enhance its persuasive ability. Furthermore, while the paper discusses differential privacy, it fails to conduct privacy attack experiments to substantiate the need for this mechanism. Additionally, the paper lacks a comprehensive experimental analysis.
4、This paper resembles a scientific report more than an academic publication. It lacks standardization in language, contains several writing errors ( line 126 ), and fails to provide a detailed description of the proposed framework.
5、The paper does not align well with the conference theme and fails to address the domains of representation learning and deep learning.

**Questions:**

Please see the weaknesses section.

---

### Official Review · Reviewer_uNUo · 2024-10-31

**Soundness:** 2
**Presentation:** 2
**Contribution:** 2
**Rating:** 3
**Confidence:** 5

**Summary:**

This paper presents a Feature-Aware Federated Learning framework for anomaly detection in 5G networks, addressing challenges of data privacy and scalability. The framework incorporates feature importance into model aggregation using integrated gradients to compute feature relevance, with differential privacy to protect client data. Additionally, a Dynamic Feature Importance Adaptation (DFIA) mechanism periodically updates feature importance to adapt to changing data distributions. Experiments show that this approach outperforms traditional federated methods like FedAvg and FedProx in unsupervised anomaly detection tasks, achieving improved accuracy, robustness, and privacy preservation.

**Strengths:**

Strengths:
+ This work offers a Feature-Aware Federated Learning framework for anomaly detection in 5G networks, addressing challenges of data privacy and scalability.

**Weaknesses:**

Weaknesses:
- The novelty of this paper needs to be further improved.
- The experimental results of this paper are not convincing.
- More STOA baselines need to be included.
- The writing quality of this paper needs another round of polishing.

**Questions:**

- The writing style of this paper suggests it may have been generated by an LLM, as it includes many informal expressions that are uncommon in academic manuscripts. It is recommended that the authors thoroughly review the language and expressions used throughout the text.

- The novelty of this paper needs further strengthening. Currently, the approach mainly combines existing techniques—feature-aware aggregation and differential privacy—without a clear distinction from previous works. The authors should further elaborate on the differences and connections between their proposed method and related work to emphasize its unique contributions.

- The paper should include more advanced baselines. In the experimental results, the authors compare only with FedProx and omit comparisons with existing advanced FL-based anomaly detection baselines, which limits the ability to demonstrate the superiority of the proposed approach.

- The accuracy of the experimental results needs further explanation. Introducing DP mechanisms typically results in a decrease in model accuracy, yet the results in this paper do not reflect this and even exceed FedAvg. Additionally, the paper should provide a detailed description of experimental settings, including learning rate, dataset size, training rounds, and other hyperparameters. Ablation experiments should be conducted to clarify the contribution of each individual component, which will aid readers in understanding the impact of each on the method's performance.

- Finally, the experimental results lack persuasiveness in terms of baselines, dataset size, dataset configurations, and parameter sensitivity analysis. The authors should consider conducting additional experiments to enhance the credibility of the experimental findings.

---

### Note · Authors · 2025-01-09

**Comment:**

Hi,

We deeply appreciate you taking the time to review our manuscript and share your thoughtful feedback. Your observations have provided valuable guidance on improving the clarity, relevance, and impact of our work.

After reviewing. on your comments, we have decided to withdraw the paper to address your suggestions more thoroughly. This will allow us to refine our research and better highlight the significance of our work.

Thank you again for your time and for helping us strengthen our work.

**Withdrawal Confirmation:**

I have read and agree with the venue's withdrawal policy on behalf of myself and my co-authors.